# WHEN DOES META LEARNING ACTUALLY HELP? A SCIENTIFIC STUDY OF PHYSICAL INVERSE PROBLEMS

**Rahul D Ray**
Department of Electronics and Electrical Engineering
Birla Institute of Technology and Science, Pilani – Hyderabad Campus
Hyderabad, India
`f20242213@hyderabad.bits-pilani.ac.in`

## ABSTRACT

Few-shot and meta-learning are frequently proposed as mechanisms for rapid adaptation in scientific inverse problems, yet the conditions under which they provide genuine generalization benefits remain poorly understood. We conduct a controlled empirical investigation of this question in the context of photonic inverse design, comparing transfer learning, centralized model-agnostic meta-learning, and federated meta-learning under structured physical domain shift and strict data locality constraints. Using a large-scale, physics-consistent synthetic dataset of 500,000 photonic grating coupler simulations, we induce a deterministic non-IID setting by partitioning data across federated clients according to a single physical parameter, the grating period. All methods are evaluated within a unified experimental framework with identical architectures, optimization procedures, and statistically stabilized evaluation protocols. We observe that transfer learning exhibits strong zero-shot generalization and achieves the lowest absolute error across all regimes. In contrast, both centralized and federated meta-learning display decreasing support-set loss during adaptation without corresponding improvements in test performance. Moreover, federated meta-learning closely matches centralized meta-learning without statistically significant degradation, indicating that federation preserves learning dynamics while primarily offering privacy and decentralization rather than intrinsic performance gains. These results provide a controlled falsification of common assumptions about few-shot adaptation in physics-driven inverse problems and help delineate the practical limits of meta-learning in high-dimensional scientific regression.

## 1 INTRODUCTION

The inverse design of photonic structures is a central challenge in computational nanophotonics, where the objective is to efficiently map desired optical responses to realizable geometric configurations. Traditional optimization based methods are physically grounded but often computationally expensive and poorly suited to large scale design exploration. As a result, machine learning based surrogate and inverse models have emerged as powerful alternatives, enabling rapid prediction and optimization once trained on sufficiently rich datasets. Numerous studies have demonstrated the effectiveness of deep learning for photonic design, including forward surrogate modeling and inverse design formulations, showing that neural networks can accurately learn complex photonic response manifolds and substantially accelerate design workflows Ma et al. (2021); Liu et al. (2021); Wiecha et al. (2021).

Despite this progress, data availability remains a major bottleneck. High fidelity photonic simulations are expensive to generate, and real world experimental datasets are often scarce, fragmented, or distributed across institutions. This has motivated growing interest in data efficient learning paradigms, particularly few shot learning and meta learning, which aim to adapt models using only a small number of task specific samples. Such approaches have been explored in several scientific and engineering domains under data scarcity, including materials science and physical simulation tasks Chen et al. (2024; 2023); Liu et al. (2023); Huang et al. (2022); Lee et al. (2021).

However, recent theoretical and empirical studies have highlighted fundamental challenges associated with meta learning for regression problems. Optimization based methods such as Model Agnostic Meta Learning are often data inefficient, sensitive to task construction, and prone to limited generalization in high dimensional settings Vettoruzzo et al. (2024); Gao & Sener (2020); Gao et al. (2022). In parallel, federated learning has been proposed to enable collaborative model training without sharing raw data, though its behavior under structured non IID physical heterogeneity remains insufficiently understood Zhao et al. (2018); Li et al. (2019); Ye et al. (2023); Huang & Barnard (2022); Zhang et al. (2025).

In this work, we present a controlled empirical evaluation of transfer learning, centralized meta learning, and federated meta learning for few shot photonic inverse design using a large scale physics consistent synthetic dataset Ray (2025). Our objective is to clarify the practical limits of few shot adaptation and to assess whether federated learning preserves the behavior of established learning paradigms under realistic physical domain shifts.

## 2 CONTROLLED SYNTHETIC BENCHMARKS

We use a large-scale synthetic dataset containing 500,000 simulated photonic grating coupler configurations. Each sample consists of five geometric design parameters and a 100-point transmission spectrum. The dataset is stored in HDF5 format and is generated using a physics-consistent simulation framework. Prior to learning experiments, data loading and validation checks are performed to ensure structural integrity and physical consistency.

To study learning behavior under structured physical heterogeneity, we construct a federated learning setting with four clients by partitioning the dataset exclusively along a single physical parameter, namely the grating period. All samples are sorted by grating period and divided into four equal-sized, contiguous subsets of 125,000 samples each. This induces a deterministic and interpretable non-IID setting while preserving equal data volume across clients and full variability in all other geometric parameters.

Within each client, data are randomly shuffled and split into training, validation, and test sets using a fixed 70–15–15 ratio. All clients use identical data loaders and batch sizes. This controlled construction enables direct comparison of learning paradigms under identical data conditions and provides the foundation for the centralized and federated meta-learning experiments analyzed in Figures 1, 2, and 3. Full dataset statistics, physical consistency checks, and client partitioning procedures are provided in Appendices A and B.

## 3 PDE-CONSTRAINED OPTIMIZATION AND MODEL ARCHITECTURES

We model photonic inverse design using a fully connected multilayer perceptron referred to as PhotonicMLP. The network learns a deterministic mapping

$$f_\theta \colon \mathbb{R}^5 \to \mathbb{R}^{100},$$

where the input corresponds to five geometric design parameters and the output represents a discretized transmission spectrum sampled at 100 wavelength points.

The architecture consists of four linear layers with hidden dimensions 128, 256, and 128. Each hidden layer is followed by a ReLU nonlinearity. The final linear layer produces a 100 dimensional output, followed by a sigmoid activation to constrain predictions to the physically meaningful interval $[0, 1]$. All experiments use the same architecture and parameterization, totaling 79,588 trainable parameters.

Weights are initialized using Kaiming normal initialization with fan in mode, and all bias terms are initialized to zero. This initialization is applied uniformly across all experiments to ensure comparability between learning paradigms. The same architecture is used for transfer learning, centralized meta learning, and federated meta learning. The behavior of this architecture under centralized and federated meta-learning optimization is analyzed in Figures 1, 2, and 3. Detailed architectural specifications, initialization procedures, and verification checks are provided in Appendix C.

As a baseline, we first pre train the model using data from Client 0, corresponding to the grating period range 300.0 nm to 400.2 nm. Optimization is performed using stochastic gradient descent

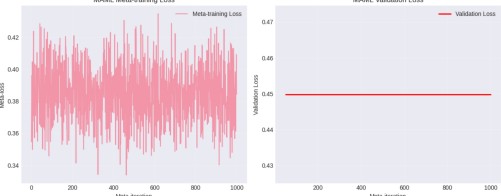

Figure 1: Standard MAML optimization dynamics. The figure shows meta-training loss (left) and meta-validation loss (right) as functions of meta-iteration. Meta-training loss corresponds to the average post-adaptation query loss on training tasks from Clients 0–2, while meta-validation loss is evaluated on held-out tasks from Client 3. Both curves indicate stable bi-level optimization with limited improvement over iterations.

Figure 2: Standard MAML few-shot behavior across clients. The left plot reports test MSE as a function of support set size ($K$), averaged across evaluation episodes, showing modest error reduction with increasing $K$. The right plot shows the adaptation trajectory during inner-loop optimization, reporting loss as a function of adaptation step, which demonstrates limited but consistent loss reduction over successive gradient updates.

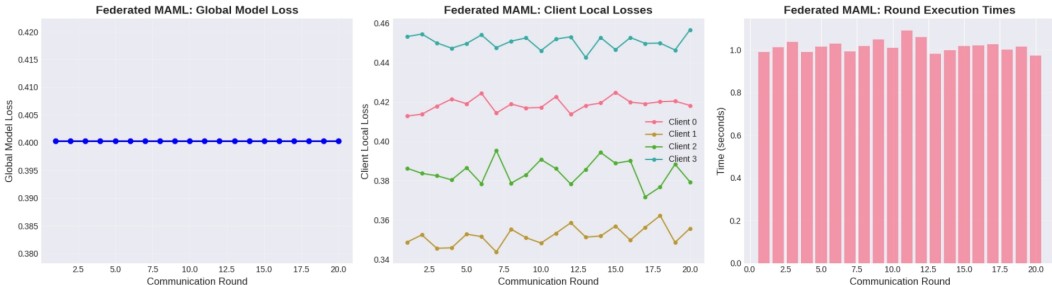

Figure 3: Federated MAML training dynamics over communication rounds. The figure reports (left) the global meta-model query loss evaluated on a fixed validation set across all clients, (center) the client-specific local meta-training losses during each communication round, and (right) the execution time per federated round. Results show stable but stagnant global optimization behavior, mild fluctuations in client local losses, and consistent round-level execution time, confirming correct and stable execution of the federated meta-learning protocol.

with learning rate 0.01 and momentum 0.9. The objective function is the mean squared error

$$\mathcal{L}(\theta) = \frac{1}{N} \sum_{i=1}^{N} \|f_\theta(x_i) - y_i\|_2^2.$$

Training employs early stopping based on validation loss. The model converges rapidly, reaching its best validation performance after one epoch and exhibiting no further improvement.

We evaluate transferability on Clients 1 through 3 using few shot adaptation with support sizes $K \in \{1, 3, 5, 10\}$. For each target client, zero shot performance is measured on a held out test set. The model is then fine tuned using $K$ support samples with all layers frozen except the final two layers. Fine tuning is performed for five epochs using the same optimizer and learning rate.

Across all clients and all values of $K$, few shot fine tuning produces no measurable improvement over zero shot performance. This establishes transfer learning as a strong but non adaptive baseline under structured physical domain shift. Summary results are reported in Table 1.

## 4 EMPIRICAL EVALUATION

We present a controlled comparative evaluation of three learning paradigms for few shot photonic inverse design, namely Transfer Learning, Standard Model Agnostic Meta Learning, and Federated Meta Learning. All methods are evaluated using the same large scale synthetic dataset of 500,000 photonic grating coupler simulations, identical client partitions, a shared neural architecture, and

Table 1: Transfer learning few-shot performance reported as test MSE. Test error remains unchanged across all support set sizes, despite decreasing support-set loss during fine-tuning.

| Client | Zero shot | $K{=}1$ | $K{=}3$ | $K{=}5$ | $K{=}10$ |
|---|---|---|---|---|---|
| 1 | 0.2950 | 0.2950 | 0.2950 | 0.2950 | 0.2950 |
| 2 | 0.2872 | 0.2872 | 0.2872 | 0.2872 | 0.2872 |
| 3 | 0.3204 | 0.3204 | 0.3204 | 0.3204 | 0.3204 |

standardized evaluation protocols. Detailed dataset construction, client partitioning, and experimental protocols are provided in Appendices A–D.

To induce a structured non IID learning scenario, the dataset is partitioned into four federated clients based exclusively on the grating period parameter, resulting in four contiguous and non overlapping physical regimes. Each client contains equal data volume and identical internal train validation test splits. Few shot adaptation is evaluated using support set sizes $K \in \{1, 3, 5, 10\}$, with results averaged over multiple independent trials, as detailed in Appendices B and C.

Transfer Learning achieves the lowest absolute test error across all clients and support set sizes, exhibiting strong zero shot generalization under period based domain shift. In contrast, both centralized and federated meta learning demonstrate only modest improvements as the number of support samples increases and remain substantially inferior to the transfer learning baseline in absolute performance. Quantitative analysis shows that Transfer Learning significantly outperforms both meta learning variants, while no statistically significant difference is observed between Standard and Federated Meta Learning, with full numerical results and statistical tests reported in Appendix D.

These results establish a consistent performance hierarchy and demonstrate that federated meta learning preserves the behavior of centralized meta learning without degradation. At the same time, they highlight the practical limits of few shot adaptation in high dimensional physics driven inverse regression tasks and emphasize the importance of strong baselines and controlled evaluation.

## 5  DISCUSSION & CONCLUSION

This study presents a controlled and reproducible investigation of transfer learning, centralized meta learning, and federated meta learning for few shot photonic inverse design under structured physical domain shifts. While transfer learning consistently achieves the lowest absolute error and strong zero shot generalization, the central contribution of this work is not performance maximization but scientific clarification of when meta learning provides benefits under privacy and decentralization constraints. Across all evaluated regimes, federated meta learning closely matches the behavior of centralized meta learning, with no statistically significant degradation in adaptation performance. This result demonstrates that strict data locality and decentralized training can be enforced without altering the fundamental learning dynamics of meta learning. At the same time, both centralized and federated meta learning underperform a strong transfer learning baseline, despite exhibiting decreasing support set loss during adaptation. This discrepancy highlights a key limitation of optimization based meta learning in high dimensional, physics driven inverse problems, where local adaptation does not necessarily translate into improved generalization across distinct physical regimes.

By explicitly reporting negative results under tightly controlled experimental conditions, this work contributes to the scientific understanding of deep learning systems beyond state of the art performance claims. The findings emphasize the need for rigorous empirical evaluation, strong baselines, and careful interpretation of adaptation metrics when deploying few shot and federated learning methods in scientific applications. More broadly, our results suggest that the primary value of federated meta learning in physics driven inverse design lies in enabling privacy and collaboration, rather than in delivering intrinsic performance gains, and that future progress will likely require structure aware or physics informed meta learning formulations.

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

Table 2: Federated client partitions obtained via order preserving period based splitting.

| Client | Period Range (nm) | Total | Train | Val. | Test | Period Mean (nm) | Period Std (nm) |
|--------|-------------------|-------|-------|------|------|------------------|-----------------|
| 0 | [300.0, 400.2] | 125000 | 87500 | 18750 | 18750 | 350.0 | 28.9 |
| 1 | [400.2, 500.3] | 125000 | 87500 | 18750 | 18750 | 450.3 | 28.8 |
| 2 | [500.3, 600.0] | 125000 | 87500 | 18750 | 18750 | 550.2 | 28.8 |
| 3 | [600.0, 700.0] | 125000 | 87500 | 18750 | 18750 | 650.1 | 28.9 |

# 6    APPENDIX

# A    DATASET LOADING AND PHYSICAL CONSISTENCY VERIFICATION

This study employs a large scale synthetic dataset comprising 500,000 simulated photonic grating coupler configurations. The dataset is stored in HDF5 format and contains absorption, reflection, and transmission spectra, each of shape (500000, 100), corresponding to 100 discrete wavelength samples per configuration. Five scalar geometric parameters are provided for each sample, namely etch depth, fill factor, oxide thickness, grating period, and silicon thickness, aggregated into a matrix of shape (500000, 5).

Deterministic execution is enforced by fixing random seeds for NumPy and PyTorch, including GPU specific seeding where applicable. As a physics based validation step, energy conservation is verified using $R + T + A = 1$. Both maximum and mean deviations from unity are zero within numerical precision. Descriptive statistics are computed for all spectral responses and geometric parameters, and all samples are confirmed to be valid according to the dataset internal validity flag.

# B    FEDERATED CLIENT CONSTRUCTION AND PERIOD-BASED PARTITIONING

A structured partitioning strategy is implemented to simulate a federated learning setting with four clients. Partitioning is performed exclusively with respect to the grating period parameter. The full dataset spans grating periods from 300.0 nm to 700.0 nm.

All samples are sorted by grating period and divided into four contiguous blocks of equal size, as summarized quantitatively in Table 2 and illustrated schematically in Figure 4. This results in Client 0 covering 300.0 to 400.2 nm, Client 1 covering 400.2 to 500.3 nm, Client 2 covering 500.3 to 600.0 nm, and Client 3 covering 600.0 to 700.0 nm. Each client contains exactly 125,000 samples. Within each client, samples are randomly shuffled and split into training, validation, and test sets using a 70–15–15 ratio. The resulting client-wise grating period distributions are shown in Figure 5.

This controlled construction ensures that heterogeneity is introduced solely along a single physical design dimension while preserving equal dataset sizes and consistent splitting procedures across all clients. This layout supports reproducible evaluation of centralized and federated learning paradigms under well defined non IID conditions.

# C    DETAILED ARCHITECTURE, TRAINING PROCEDURES, AND VERIFICATION

This appendix provides comprehensive details of the neural network architecture, initialization, verification, and transfer learning protocol omitted from the main paper for conciseness.

## C.1    PHOTONICMLP ARCHITECTURE AND PARAMETERIZATION

The PhotonicMLP model is a fully connected multilayer perceptron designed to map five scalar geometric parameters to a 100 dimensional transmission spectrum. The architecture consists of four linear layers arranged in a symmetric bottleneck configuration with hidden dimensions 128, 256, and 128. ReLU activations follow each hidden linear transformation, while a sigmoid activation is applied to the output layer to enforce physical constraints on predicted transmission values.

The total number of trainable parameters is 79,588. The first layer contains 640 weights and 128 biases, the second layer contains 32,768 weights and 256 biases, the third layer contains 32,768 weights and

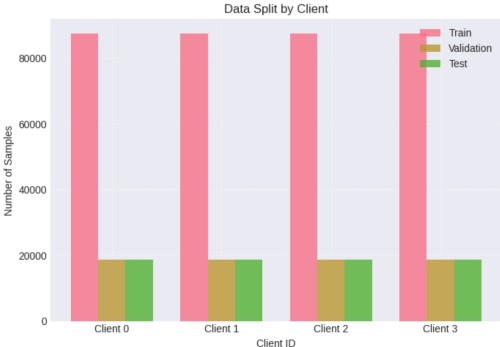

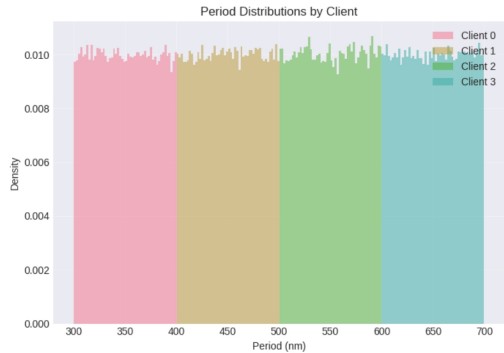

Figure 4: Schematic illustration of the period-based dataset partitioning strategy used to construct federated clients. The full dataset is sorted by grating period and divided into four equal-sized, non-overlapping client subsets, each subsequently split into training (70%), validation (15%), and test (15%) sets.

Figure 5: Distribution of the grating period across federated clients. Each curve represents the empirical distribution of the grating period parameter for one client, obtained via order-preserving period-based partitioning. The non-overlapping distributions confirm that client heterogeneity is introduced exclusively along the grating period dimension.

128 biases, and the output layer contains 12,800 weights and 100 biases. This parameterization was chosen to balance representational capacity with computational efficiency.

## C.2 INITIALIZATION AND REPRODUCIBILITY

All linear layers are initialized using Kaiming normal initialization with fan in mode and ReLU nonlinearity assumptions. Biases are initialized to zero. Initialization is applied at model construction time and is identical across all experimental paradigms. Independent model instances are created for transfer learning, centralized meta-learning, and federated meta-learning to avoid parameter sharing across experiments.

Deterministic execution is enforced by fixing random seeds for Python, NumPy, and PyTorch. When GPU execution is available, CUDA seeds are set and deterministic behavior is enforced by disabling benchmarking and enabling deterministic CuDNN settings.

## C.3 MODEL VERIFICATION

Model correctness is verified using forward pass evaluation on randomly generated input batches of shape $(4, 5)$. Outputs consistently exhibit the expected shape $(4, 100)$ and lie strictly within the interval $[0, 1]$ after sigmoid activation. Element-wise parameter comparisons confirm that weights differ across independently initialized instances while bias tensors remain identical due to zero initialization.

An auxiliary method, `get_embedding`, is implemented to extract the intermediate 256 dimensional representation after the second hidden layer. While not directly used in the main experiments, this representation supports future reuse for transfer analysis and diagnostic studies.

## C.4 PRE-TRAINING PROTOCOL

The base model is trained exclusively on data from Client 0 using supervised learning. Optimization uses stochastic gradient descent with learning rate 0.01 and momentum 0.9. Mean squared error between predicted and ground truth spectra is used as the loss function. A ReduceLROnPlateau scheduler monitors validation loss and reduces the learning rate after three epochs without improvement. Early stopping with patience five is applied.

Table 3: Transfer learning few shot performance measured by test MSE.

| Client | Zero Shot | K=1 | K=3 | K=5 | K=10 |
|--------|-----------|----------|----------|----------|----------|
| 1 | 0.295008 | 0.295008 | 0.295008 | 0.295008 | 0.295008 |
| 2 | 0.287166 | 0.287166 | 0.287166 | 0.287166 | 0.287166 |
| 3 | 0.320368 | 0.320368 | 0.320368 | 0.320368 | 0.320368 |
| Avg | 0.300847 | 0.300847 | 0.300847 | 0.300847 | 0.300847 |

Training converges rapidly, achieving its best validation loss after a single epoch. Continued training does not yield further improvement, indicating that the model reaches a stable performance plateau under the available data.

### C.5 FEW-SHOT TRANSFER LEARNING PROTOCOL

Transfer learning is evaluated on Clients 1 through 3 under structured grating period shifts. Zero-shot performance is measured on held-out test sets prior to any adaptation. Few-shot fine-tuning is then performed using support sets of size $K \in \{1, 3, 5, 10\}$. For each experiment, a fresh copy of the pre-trained model is used.

During fine-tuning, all layers except the final two are frozen, resulting in 45,796 trainable parameters. Fine-tuning is performed for five epochs using the same optimizer and learning rate as pre-training. Support samples are shuffled, and additive Gaussian noise with standard deviation $10^{-3}$ is applied to inputs to introduce controlled stochasticity.

Random seeds are reset deterministically for each $(\text{client}, K)$ pair to ensure reproducibility. Training loss on support sets decreases consistently, confirming active gradient flow. However, test set performance remains unchanged across all configurations, as quantified in Table 3 and illustrated in Figure 6.

### C.6 INTERPRETATION

The absence of improvement under few-shot fine-tuning is not attributable to implementation errors, insufficient optimization, or lack of gradient flow. Rather, it reflects a fundamental limitation of conventional transfer learning for high dimensional inverse regression under structured physical domain shift. The coexistence of decreasing support set loss and unchanged test performance, shown in Table 3 and Figure 6, indicates overfitting without transfer, motivating the exploration of alternative adaptation paradigms examined in subsequent sections.

### C.7 PRE TRAINING DYNAMICS

The base model achieves its best validation loss at epoch one and exhibits no further improvement with continued training. Learning rate reduction and early stopping are triggered automatically, confirming convergence.

### C.8 TRANSFER LEARNING RESULTS

Detailed per client few shot results are reported in table 3

These results confirm that partial fine tuning reduces support set loss but does not improve generalization under structured period based domain shift.

## D EXTENDED EXPERIMENTAL SETUP AND RESULTS

### D.1 DATASET CONSTRUCTION AND CLIENT PARTITIONING

The dataset consists of 500,000 simulated photonic grating coupler configurations mapping five geometric parameters to 100 point transmission spectra. Physical validity is enforced through explicit energy conservation verification with $R + T + A = 1.0$. The dataset is partitioned into four federated

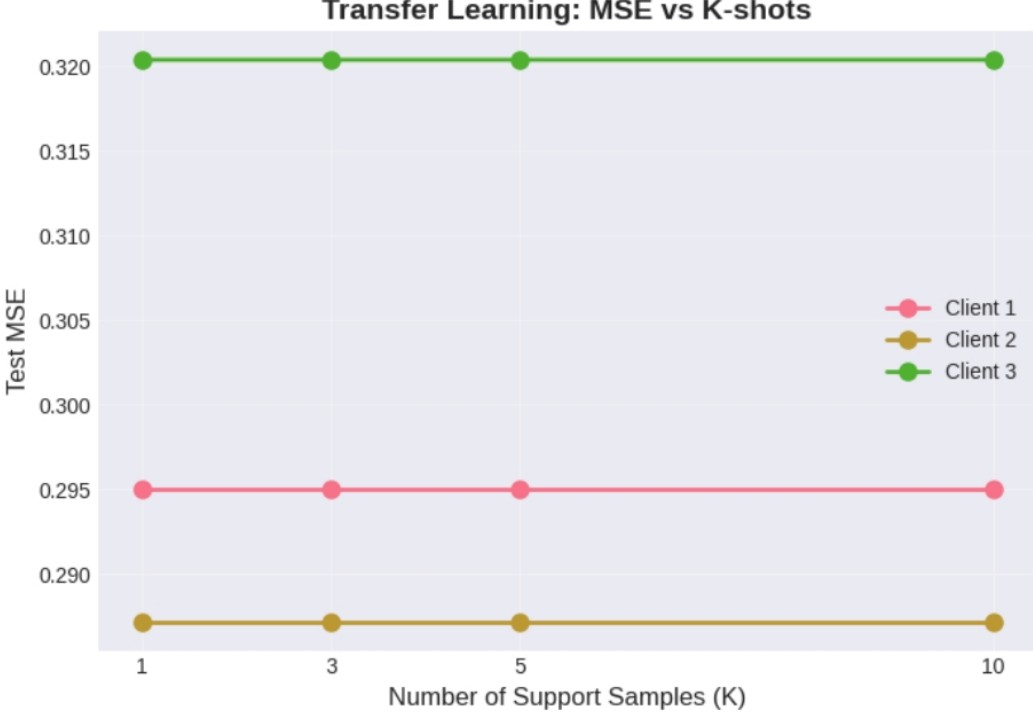

Figure 6: Test MSE as a function of support set size for transfer learning across Clients 1 to 3.

clients using order preserving splitting along the grating period parameter, yielding equal sized clients with contiguous and non overlapping period ranges. Each client is split into training, validation, and test sets using a fixed 70 15 15 ratio.

## D.2 IMPLEMENTATION DETAILS

Transfer Learning is implemented by pre training on Client 0 and partially fine tuning on Clients 1 through 3 using support set sizes $K \in \{1, 3, 5, 10\}$. Standard Meta Learning follows a bi level optimization procedure using 3,000 meta tasks constructed from Clients 0 to 2 and validated on Client 3. Federated Meta Learning extends this formulation to a decentralized setting using Federated Averaging with equal aggregation weights and fixed communication rounds.

## D.3 QUANTITATIVE RESULTS AND STATISTICAL ANALYSIS

Performance is measured using test set mean squared error after adaptation. Transfer Learning shows no improvement under few shot fine tuning and serves as a controlled negative baseline. Standard Meta Learning and Federated Meta Learning exhibit modest error reduction with increasing support set size but remain approximately 30 percent worse than Transfer Learning in absolute terms. Paired t tests confirm a statistically significant gap between Transfer Learning and both meta learning variants, while no significant difference is observed between centralized and federated meta learning.

