# OpenReview forum: "WHEN DOES META LEARNING ACTUALLY HELP? A SCIENTIFIC STUDY OF PHYSICAL INVERSE PROBLEMS"
_ICLR.cc/2026/Workshop/Sci4DL — Sci4DL 2026_

### Official Review · Reviewer_WCvx · 2026-02-14

**Fit:** 2
**Significance:** 2
**Confidence:** 2

**Summary:**

This paper empirically studies when meta-learning helps by benchmarking centralized MAML and federated MAML against a strong transfer-learning baseline on a physics-simulated photonic grating-coupler inverse design regression task. Using a controlled non-IID split across clients, authors find that transfer learning achieves the best test MSE, while MAML-style adaptation often reduces support-set loss without improving test performance; federated MAML closely matches centralized MAML, suggesting federation mainly preserves privacy/locality rather than boosting accuracy.

**Strengths:**

This paper uses a controlled photonic inverse-design regression benchmark (non-IID client splits by grating period) to test when meta-learning helps, comparing centralized MAML and federated MAML against a strong transfer-learning baseline; empirically, transfer learning achieves the best test MSE, while MAML often shows the misleading pattern that support-set loss decreases during adaptation without improving test performance, and federated MAML closely tracks centralized MAML, suggesting federation mainly preserves privacy/data locality rather than boosting accuracy.

**Suggestions:**

I’m not familiar with this specific photonic task and dataset, so I’d suggest adding a more standardized evaluation protocol to significantly broaden the paper’s audience and make the conclusions easier to compare and trust.

---

### Official Review · Reviewer_U6ee · 2026-02-16

**Fit:** 3
**Significance:** 2
**Confidence:** 3

**Summary:**

This paper tests whether meta-learning improves few-shot performance in photonic inverse design under structured non-IID domain shift. Using a controlled 500k-sample dataset, it compares different methods with identical setups. Transfer learning achieves the lowest test MSE, while both meta-learning variants fail to improve generalization despite lowering adaptation loss. Federated MAML matches centralized MAML without degradation.

**Strengths:**

- The structured non-IID client construction along a single physical parameter (grating period) is scientifically interpretable and enables clear analysis of domain shift effects

- The work makes a valuable negative result contribution by demonstrating that decreasing adaptation loss does not necessarily translate to improved generalization in physics-driven regression.

**Suggestions:**

- The task is limited to a single synthetic photonic inverse design benchmark with one specific architecture (MLP). It is unclear whether conclusions generalize to other physical systems and architectures (e.g., GNNs, transformers).

- Meta-learning is evaluated primarily through MAML-style optimization; alternative formulations (e.g., representation-based meta-learning or physics-informed priors) are not explored, which may limit the scope of the claim.

---

### Official Review · Reviewer_4JDY · 2026-02-22

**Fit:** 1
**Significance:** 1
**Confidence:** 2

**Summary:**

The paper shows that transfer learning exhibits strong performance across other methods.

**Strengths:**

The implementation details are clear.

**Suggestions:**

- The paper seems to focus on photonic grating coupler simulations only, instead of general physical inverse problem, which is inconsistent with the title.
- Figure 1, 2, 3 is unclear, it is better if the authors could give more analysis and justifications on why those methods perform badly. Also Figure 1, 2(a) seems to be performed on 4 clients only.
- Implementation details in section 2, 3 can be put in appendix, more analysis or experiments are needed
- It is also unclear what assumptions in physics-driven inverse problem is falsified.

---

### Meta-Review · Area_Chair_FhNw · 2026-03-01

**Recommendation:** Accept

**Metareview:**

All reviewers indicate that the choice of photonic inverse design setting is too limiting to the scope of the results. However, two reviewers find the results fairly significant because the paper on shedding light on comparison between transfer learning, meta learning and federated meta learning for few shot learning (albeit in a limited setting).

---

### Decision · Program_Chairs · 2026-03-02

Accept